# The Importance of Behavioral and Native Factors on COVID-19 Infection and Severity: Insights from a Preliminary Cross-Sectional Study

**DOI:** 10.3390/healthcare10071341

**Published:** 2022-07-19

**Authors:** Hani Amir Aouissi, Mohamed Seif Allah Kechebar, Mostefa Ababsa, Rabih Roufayel, Bilel Neji, Alexandru-Ionut Petrisor, Ahmed Hamimes, Loïc Epelboin, Norio Ohmagari

**Affiliations:** 1Scientific and Technical Research Center on Arid Regions (CRSTRA), Biskra 07000, Algeria; m.kechebar@mesrs.dz (M.S.A.K.); mostefa.ababsa@gmail.com (M.A.); 2Laboratoire de Recherche et d’Etude en Aménagement et Urbanisme (LREAU), Université des Sciences et de la Technologie (USTHB), Algiers 16000, Algeria; 3Environmental Research Center (CRE), Badji-Mokhtar Annaba University, Annaba 23000, Algeria; 4College of Engineering and Technology, American University of the Middle East, Kuwait; bilel.neji@aum.edu.kw; 5Doctoral School of Urban Planning, Ion Mincu University of Architecture and Urbanism, 010014 Bucharest, Romania; alexandru.petrisor@uauim.ro; 6National Institute for Research and Development in Tourism, 50741 Bucharest, Romania; 7National Institute for Research and Development in Constructions, Urbanism and Sustainable Spatial Development URBAN-INCERC, 021652 Bucharest, Romania; 8Faculty of Medicine, University Salah Boubnider of Constantine 3, Constantine 25000, Algeria; ahmed.hamimes@univ-constantine3.dz; 9Infectious and Tropical Diseases Department, Centre Hospitalier de Cayenne Andrée Rosemon, 97306 Cayenne, France; loic.epelboin@ch-cayenne.fr; 10Centre d’Investigation Clinique (CIC INSERM 1424), Centre Hospitalier de Cayenne Andrée Rosemon, 97306 Cayenne, France; 11Disease Control and Prevention Center, National Center for Global Health and Medicine, Tokyo 162-8655, Japan; nohmagari@hosp.ncgm.go.jp; 12AMR Clinical Reference Center, National Center for Global Health and Medicine, Tokyo 162-8655, Japan

**Keywords:** COVID-19, public health, preventive measures, behavior, infectious diseases

## Abstract

The COVID-19 pandemic has had a major impact on a global scale. Understanding the innate and lifestyle-related factors influencing the rate and severity of COVID-19 is important for making evidence-based recommendations. This cross-sectional study aims at establishing a potential relationship between human characteristics and vulnerability/resistance to SARS-CoV-2. We hypothesize that the impact of the virus is not the same due to cultural and ethnic differences. A cross-sectional study was performed using an online questionnaire. The methodology included the development of a multi-language survey, expert evaluation, and data analysis. Data were collected using a 13-item pre-tested questionnaire based on a literature review between 9 December 2020 and 21 July 2021. Data were statistically analyzed using logistic regression. For a total of 1125 respondents, 332 (29.5%) were COVID-19 positive; among them, 130 (11.5%) required home-based treatment, and 14 (1.2%) intensive care. The significant and most influential factors on infection included age, physical activity, and health status (*p* < 0.05), i.e., better physical activity and better health status significantly reduced the possibility of infection, while older age significantly increased it. The severity of infection was negatively associated with the acceptance (adherence and respect) of preventive measures and positively associated with tobacco (*p* < 0.05), i.e., smoking regularly significantly increases the severity of COVID-19 infection. This suggests the importance of behavioral factors compared to innate ones. Apparently, individual behavior is mainly responsible for the spread of the virus. Therefore, adopting a healthy lifestyle and scrupulously observing preventive measures, including vaccination, would greatly limit the probability of infection and prevent the development of severe COVID-19.

## 1. Introduction

### 1.1. Background

According to the WHO (World Health Organization), several coronaviruses (CoVs) have been reported since 2002. The coronavirus responsible for the 2019 pandemic was named SARS-CoV-2 by the ICTV (International Committee on Taxonomy of Viruses) due to its similarity to SARS viruses [1].

At the end of 2019, a novel beta coronavirus was identified in China’s Hubei province, in Wuhan [2]. By that time, the virus had spread and disrupted all aspects of human life worldwide. The symptoms of severe acute respiratory syndrome coronavirus 2 are similar to those of previously known coronavirus infections, including fever, dry cough, and fatigue; however, the SARS-CoV-2 has higher transmissibility [3,4]. It has largely surpassed MERS and SARS in terms of both the spatial range of epidemic areas and the number of infections. The global health crisis experienced has posed a significant threat to public health [5].

According to the WHO’s monthly operational update on COVID-19 of July 2022, there are almost 552 million confirmed cases and 6.34 million related deaths worldwide [6]. The consequences were serious, especially at the start of the epidemic, given the lack of information and means to fight this virus [7]. Across the world, a significant number of proactive measures were adopted, including distance education, banning international traveling, and campaigns encouraging everyone to “stay at home” [8]. Given the unknown and unexpected aspects of this epidemic in modern times, there is inescapably a lack of research on its psychosocial effects [9]. 

Thus, understanding the innate and lifestyle-related factors influencing the incidence and severity of COVID-19 is critical for making appropriate recommendations to prevent the transmission of COVID-19 as well as the development of severe COVID-19. However, the detailed characteristics influencing the rate and severity of COVID-19 are not fully understood. Contrary to previously published studies, which were mostly concerned with the influence of COVID-19 on changes in lifestyle, the present study is mainly interested in the influence of lifestyle on COVID-19 infection [10,11]. So far, COVID-19 questionnaires have been much more related to the perception of people concerning this epidemic [12]. Since the epidemic’s occurrence, several international questionnaires have been created. Their purpose was to answer a specific question, e.g., about the knowledge and attitudes of residents in the prevention and control of COVID-19 [2] or the preventive practices against the COVID-19 pandemic in the general population [13]. Moreover, they were about the mental health impact on people with and without depressive, anxiety, or obsessive-compulsive disorders [14], eating habits, activity, and sleep behavior [15], etc. Others focused on particular countries such as China [16], the USA [17], and France [18]. Herein, we conduct a cross-sectional analysis to evaluate the influence of innate and lifestyle characteristics on COVID-19 by using an online questionnaire based on the literature review, as follows. 

### 1.2. Literature Review on the Main Risk Factors

A comprehensive literature review using databases and search engines, such as Web of Science, Google Scholar, PubMed, and Scopus, was carried out to acquire a global overview of the relationships between the different risk factors and COVID-19. The following keywords: “coronavirus”, “COVID-19”, “questionnaire”, “survey”, “tobacco”, “alcohol”, “lifestyle”, “behavior”, “immunity”, “ethnic origin”, “continent”, “country”, “epidemic”, “blood group”, “sports activity”, “age”, “educational attainment”, and “preventive measures” were included in our search. The search yielded more than 201 related articles. Furthermore, after screening the titles, abstracts, and full contents, 160 articles were found relevant and cited in this manuscript. Out of these, 64 were used in the literature review.

The literature review allowed us to identify the risk factors described in the introduction. The conclusion of several studies has amply demonstrated that changes in individual factors would directly (or indirectly) affect the spread of the virus. Through literature review and expert opinion, some factors were found to be determinant, e.g., age, social distancing, air temperature, ventilation/airflow, humidity, population density, and community consciousness. These factors were interdependently found to have a strong impact on the SARS-CoV-2 epidemic characteristics [19,20]. The selected factors are presented in the following sections according to the probability of infection, from the highest to the lowest.

#### 1.2.1. Physical Activity

Regular sports activity is associated with a decrease of 6.0% to 9.0% of the risk of influenza-associated mortality [21]. It was demonstrated previously, through a study on mice that exercise (moderately) in the first days after an influenza virus infection, that the mortality was reduced [22]. However, it remains unclear how physical exercise affects infections. Endurance exercise that lasts less than 1 h stimulates NK cell cytotoxicity [23]. Moreover, mild physical activity may boost the immune system, while exhausting exercise may weaken it [24]. In general, an unhealthy lifestyle could have negative consequences, whether dealing with the virus now or in the post-COVID-19 period, specifically in sedentary people. The consequences that a sedentary situation can cause and which we can observe over time are, inexorably, cardiovascular diseases such as obesity, diabetes, hypertension, and metabolic syndrome because a decrease in activity is logically associated with a reduction in insulin sensitivity according to Narici et al. [25]. The same authors claimed that inactivity during confinement could result in a reduction of cardiac volume, oxygen absorption capacity, and VO_2_ max (maximum oxygen consumption). A reduction in VO_2_ max and the absorption capacity of oxygen (O_2_) is often associated with a high mortality rate. In addition, this decrease affects blood circulation and the oxidative function of muscles. In these circumstances, sports activity can be defined as a key strategy to fight against unhealthy lifestyles during the pandemic or even after [26] since it contributes to maintaining an optimal state of health (mental/physical). The WHO has proposed many general recommendations for sports activity to combat the confinement situation. It recommends performing 150 min or at least 75 min/week of physical activity (moderate or vigorous). It also recommends combining activities of different intensities [27]. 

#### 1.2.2. Age

Susceptibility to SARS-CoV-2 is universal, but older age (in addition to other factors) is associated with disease severity. Hence, since the beginning of the epidemic, age has been defined as the keystone in COVID-19-infected people [28]. The first data coming from Italy also attested that mortality was highly significant in septuagenarian patients and even higher in octogenarians [29]. A meta-analysis has suggested an essential influence of age on the mortality of infected patients, with a relevant threshold of age >50 and, especially, >60 [30]. This epidemic reminds us daily that older individuals, specifically those with rheumatic diseases [31], are extremely vulnerable to infectious diseases due to co-morbidities linked with age and a decrease in immunological competence or “immunosenescence” [32]. This process, particularly the increased production of inflammatory cytokines resulting from inflammation, is partly responsible for determining the prognosis of COVID-19 in elderly individuals [33]. 

The peculiarities of the immune system of older individuals may contribute to both the deficiency of effector mechanisms essential for fighting viral pathogens and an exacerbated inflammatory response, which can accelerate and intensify lung tissue damage [34]. Nonetheless, several authors agree on the fact that one should in no way consider COVID-19 as a “disease of the elderly” but that the pandemic should be considered a temporary setback in a long-term decline in senescent mortality [35]. 

#### 1.2.3. Resistance (Immunity)

The genetic background of patients and the presence of concomitant pathological conditions were also examined; it was provided that adaptive immunity to closely related viruses or other microorganisms can reduce susceptibility or increase the severity of disease [36]. In line with some studies, seemingly, the most severely affected people have misguided antibodies (autoantibodies) that attack the immune system rather than the virus that causes the disease. A small percentage of the population who develop severe COVID-19 is characterized by a specific genetic mutation that affects immunity. Consequently, many individuals lack effective immune responses, which depend on type I interferon [37,38].

At the same time, it seems that a part of the population is more resistant to COVID-19; for example, according to the study by Le Bert et al. [39], some people have developed immune cells called T lymphocytes, which are directed against the new coronavirus. This immune response would be “lasting”, unlike antibodies, which would disappear quickly. Above all, this response would be present in more than 50.0% of healthy people who have never been infected with the coronavirus. This also demonstrates that different efficacies of both innate and immune responses, according to age and co-morbid conditions, can be a confounding factor [40]. Grasping the combinational and individual roles of hosts, environmental factors, and viral factors in COVID-19 infection provides better knowledge of the eventual high-risk groups of people concerning SARS-CoV-2, specifically in terms of severity and susceptibility [41]. 

#### 1.2.4. The Continent of Residence

As a pandemic, COVID-19 has hit almost every country on earth. Different approaches to emergency management were adopted in each country [42], as in East Asia [43]. Some studies have analyzed the differences and similarities in how the inhabitants of different countries responded to government policies enacted during the spread of the COVID-19 pandemic [44] by measuring, for example, confirmed cases and deaths using the stringency index. The results varied according to each country [45]. If possible, a more government-related response would be worth investigating. Nevertheless, it has been demonstrated that the lag times in responding to government policy and response variability in each country are significantly correlated with the number of infections and deaths [46].

#### 1.2.5. Ethnicity

Ethnicity is a complex concept that can include multiple dimensions as it is socially constructed and may be associated with biological attributes, such as skin color [47]. Black, Asian, and minority ethnic communities in the UK made up to 36.0% of the critically ill patients with COVID-19 that required intensive care [48]. This phenomenon is not specific to the UK; it turns out that this is usually the case in several countries. There is considerable evidence that the incidence of clinically important diseases is higher in many minority ethnic groups despite the fact that impacts of race/ethnicity are likely to vary in different countries, as underlined by some studies [49,50].

Some meta-analyses were published recently. According to Sze et al. [51], there is clear confirmation that patients of Asian, Black, and Hispanic ethnicities have an increased probability of being infected with coronavirus compared to white ethnics. In addition, there is a possibly higher risk of intensive care admission and death in Asians. Their findings should be used by institutions and decision-makers to minimize exposure to COVID-19 in ethnic minorities. This can be achieved through some measures such as helping them access health care resources quickly, observing ethical standards, and eliminating as much as possible all forms of racism, social determinism, and inequalities. For Vist et al. [52], although there is an increased risk of infection and hospital admission due to SARS-CoV-2 for several immigrants and minority ethnic groups, this is especially associated with low socioeconomic status, resulting in higher admission rates to hospitals and higher death rates compared to groups with high socioeconomic status, the fact that may explain the tragic situation in some countries compared to others with almost similar ethnicities [53]. 

#### 1.2.6. Blood Group

The majority of studies reported that blood group A had an increased frequency amongst SARS-CoV-2 infection risk, with the opposite for blood group O, which is associated with a decreased frequency and risk of infection [3,54]. However, recent studies have not shown an association between blood group and COVID-19 infection susceptibility or blood group O and a decreased risk of infection [55]. The role of blood group in the infectivity of SARS-CoV-2 and COVID-19 severity requires further studies. The role of ABO type is likely secondary and non-modifiable [56], and, apart from blood groups, other population-dependent antigen distributions may be of importance. It is recognized that genetic differences contribute to variations in the immune response showed by different individuals to a pathogen. These differential responses may influence the spread of disease, indicating why such diseases impact some populations more than others. For example, it has been demonstrated that the susceptibility of individuals in terms of the HLA genotype is an important factor that can explain the potential spread of different viruses among different ethnicities and populations [57]. 

#### 1.2.7. Observance of Protective Measures

According to the WHO [6], there are many health precautions that individuals can take to avoid COVID-19 infection, including frequent handwashing, wearing facial coverings, maintaining a 2 m distance between themselves and others, staying away from crowds, coughing/sneezing into the elbow (or mask), cleaning/disinfecting surfaces regularly, and examination by a doctor if necessary, as recommended by the CDC [58]. In addition, we included vaccination among the strict preventive measures, given that when the questionnaire was created, the vaccination process was barely starting and its contribution was almost negligible. However, we have still considered it as an additional measure and discussed it in more detail in the introduction. 

#### 1.2.8. Educational Attainment

It has been demonstrated that education level has an effect on the incidence of some diseases (specifically non-communicable diseases). Low levels of education are related to the high prevalence and incidence of cerebrovascular and cardiovascular diseases, cancer, diabetes, hypertension, and chronic respiratory diseases [59]. Generally, a bad socioeconomic status may be linked to a low educational level, thus increasing the risk of previously mentioned diseases [60] and of COVID-19 because of low levels of immune cells and high levels of cytokines in body fluids [61]. It should be remembered that access to education is not the same elsewhere; when examining urban versus rural outcomes in COVID-19, the disparities are markedly evident when considering county-level data [62]. Concerning the relationship between COVID-19 and educational level, it is acknowledged that education and engagement in health behaviors are positively associated [63], and low education is associated with unhealthy behaviors [64]. However, several external factors may be examined, and additional measures will have to be studied each time to test for this relationship [65]. Some studies have also suggested that a standardized measurement should be used in future studies [66]. 

#### 1.2.9. Tobacco

According to the WHO, tobacco is one of the main causes of premature death and morbidity. It is known that smoking and consuming smokeless tobacco (SLT) products significantly increases the risk of NCDs (non-communicable diseases). According to Islam and Walton [67], only a few studies have investigated the relationship between tobacco consumption and COVID-19 [68]. Some of them have shown that the hospitalization rate of smokers is higher than that of non-smokers [69,70]. Berlin et al. [71] encouraged stopping smoking and repeated gestures that facilitate contamination and called for the awareness of health authorities. Mistry et al. [72], in their study, found an increased risk of mortality specifically for people over 65, especially those who live in urban areas, because they are exposed to pollution. Yingst et al. [68] stated that the increased stress can be detrimental, and it is imperative to create innovative methods to support users interested in quitting during this particularly difficult period. 

#### 1.2.10. Alcohol

Alcohol consumption increases the risk of community-acquired infections [73]. Nonetheless, news and media have shown that the false propaganda of racketeers misled the public into believing that, since alcohol can remove viruses, its consumption can destroy the coronaviruses after entering the body, supposedly because ethanol is commonly used for hand sanitizers [74]; however, this is nothing more than a myth. Some studies demonstrated that alcohol poisoning, which is a consequence of an increase in alcohol consumption, increased significantly with the appearance of the SARS-CoV-2 epidemic [75,76]. In most communities, alcohol poisoning is responsible for several health issues, although it can easily be avoided [77]. The availability of methanol and ethanol in recent years has led to an augmentation of morbidity and mortality due to alcohol poisoning in some low-income countries, specifically Muslim ones [78]. Lassen et al. [79] demonstrated that weekly alcohol consumption is associated with the progression to ARDS during hospitalization with COVID-19; however, there is a lack of studies dealing with alcohol consumption and infection by SARS-CoV-2. 

#### 1.2.11. Gender

Generally, data around the world indicate higher COVID-19 case fatality rates among men than women [80]. Gender-based behavioral and socio-cultural differences may contribute to the sex differences reported in the severity of COVID-19 disease. For example, men do not wash their hands as much, especially with soap or other products, after entering a restroom and are more likely to smoke [81]. 

Large-scale data from two meta-analysis studies demonstrated that although there are no significant differences in the proportion of individuals infected with the virus, men are much more likely to develop serious illnesses and die compared to women [82,83], except for some countries such as India [84]. According to Bhopal and Bhopal [85], hypotheses based on risk factors that are known to change with both sex and age seem to be the most probable explanations for the observed differences. The differences are due to lifestyle (e.g., alcohol, smoking), occupation, use of medications, or medical co-morbidities. These descriptions reflect cultural and social characteristics associated with gender rather than the biology of sex. 

### 1.3. Aims and Importance of Research

Given the ambiguity of answers from previous studies that have used questionnaires, the present research was designed to correct some past limitations. The aim of this study is to investigate the effects of factors beyond human control (e.g., blood group, ethnicity) and lifestyle or behavioral characteristics (e.g., tobacco use, sports activity, and alcohol consumption) on COVID-19 potential infection and severity. Hence, based on the fact that the number of infections varies greatly by country and continent (even if this may also be due to significant variations in the level of underreporting of cases between countries, in addition to the low number of tests in some countries), we hypothesize that the innate factors should be the most significant in deciding on the possible infection of individuals. We examined this assumption using questionnaires for quantification.

## 2. Materials and Methods

### 2.1. Overview

The present study is based on a cross-sectional design using an online questionnaire. The main function of the questionnaire was to give the survey a greater extension. It was also used as a means to verify statistically the extent of generalization that can be reached by the existent information and preliminary assumptions [86]. 

To accomplish the aim, we designed an anonymous online survey with a sufficient number of items; its qualities include conciseness, scientific structure, pre-testing, and global practicability. Considering the lack of similar works, the questionnaire addressed some basic information and all major lifestyle aspects. To ensure optimal compliance with preventive measures against the spread of the virus, we deliberately chose to share the survey exclusively online to limit contact as much as possible. A standardized methodology, including steps such as literature review, expert review, ethical approval, and pre-testing, was undertaken to develop and validate the questionnaire [87,88]. 

### 2.2. Development of the Questionnaire

The online questionnaire was prepared in Arabic, English, Spanish, and French, as these are the four native languages of the authors of the current study and are also in the top 6 most spoken languages worldwide [89]. This allows for the avoidance of oversights in the preparation of the questionnaire, the interpretation of the answers, the facility to respond to the questions, and doubts of the survey participants, if needed. The data were collected via Google Forms through a self-report questionnaire between 9 December 2020 and 21 July 2021. Additionally, the questionnaire consisted of three main parts. The first one was related to socio-demographic characteristics. The second part consisted of items related to behavior, and the last part included the infection status (yes or no) and its severity. We made the questionnaire as easy as possible, with a minimum of items, so that it could be understood by almost the entire global population. We have particularly made sure that it is very fast to answer, so as not to be discouraging, while deliberately excluding criteria that are too subjective or difficult to measure (e.g., hours of sleep, quality of nutrition, and stress rate). To highlight the importance of the research and its great credibility, the tool was presented to a committee of national and international experts who helped to improve it by deleting or rewording some items. Their intervention made it possible to ensure its suitability and validity before applying it to the participants. Conventionally, ethical approval and consent were obtained from the CRSTRA and all participants. After pre-testing, the questionnaire was first shared with the families, friends, and colleagues of the study’s authors. Then, the second step was to share the questionnaire via social media, including Instagram, Facebook, LinkedIn, and WhatsApp, and volunteer participants were requested to fill in the online form, targeting a more heterogeneous population, according to the method of Sawik and Plonka [90] and Kong et al. [91]. In fact, social media networks, especially Facebook, are now commonly used in medical research, specifically for cross-sectional design, and many studies have demonstrated its usefulness for quickly recruiting a large sample at the lowest cost [92,93,94]. Concerning other social media, although they are a little more recent than Facebook, their use is becoming more and more democratized in new studies, especially Instagram [95,96] and WhatsApp [97,98]. At the same time, we required that the respondents should be at least 18 years old to participate and asked for confirmation from each respondent. In summary, our survey was conducted following the procedure of Kühne and Zindel [99].

### 2.3. Statistical Analysis

Statistical analyses were performed using SAS^®^ (version 9.4). We coded the contamination factors on a numerical scale due to their nature (nominal or ordinal variable). For this reason, the responses were coded as follows: 

A progressive scale describing the direction of variation concerning the factors of an ordinal nature:

Educational attainment: 0–3 (0 = Not precise, 1 = No study or primary, 2 = Middle/secondary, 3 = University/post-university)—Age: 1–4 (1 = 18–30, 2 = 31–45, 3 = 46–59, 4 = 60+)—Sports activity: 1–3 (1 = Little or no activity, 2 = Moderate, 3 = Very active)—Health status: 1–3 (1 = Resistant, 2 = Moderately sensitive, 3 = Very sensitive)—Tobacco use: 1–3 (1 = No, 2 = Occasionally, 3 = Frequently)—Alcohol consumption: 1–3 (1 = No, 2 = Occasionally, 3 = Frequently)—Protective measures against COVID-19: 1–3 (1 = Not at all, 2 = Medium application, 3 = Strict application)—Severity: 0–3 (0 = Healthy, 1 = Low, 2 = Treatment, 3 = Intensive care).

A two-choice scale for the question: “Have you been affected by COVID-19?” (No = 0, Yes = 1).

A random scale for factors of a nominal nature: Residence (1 = Africa, 2 = Europe, 3 = North America, 4 = South America, 5 = Asia, 6 = Oceania)—Ethnicity (0 = Not precise, 1 = Other, 2 = African/Afro-American, 3 = Caucasian, 4 = Arabic, 5 = Asian, 6 = Latino)—Gender (0 = Not precise, 1 = Male, 2 = Female, 3 = Other).

We noticed that the estimations of odds ratios were affected by a very small sample size for specific levels of the variables. In order to increase the validity of statistical analyses, data for some levels were removed by changing them to missing observations: Gender—3 “Other” values, Continent—2 “Oceania” values, Health—1 “0 = Healthy” value, Continent—10 “4 = South America” values, and Ethnicity—11 “6 = Latino” values. However, we did not eliminate any blood groups, despite the reduced sample size influencing, in one case, the estimation of odds ratios because it makes no sense from a health standpoint.

Two statistical analyses were run:The dependence of having been affected before or not by COVID-19 on the potential risk factors investigated was analyzed using logistic regression, a statistical method modeling a binary event as the probability of its occurrence.The dependence of the severity of infection on potential risk factors investigated was also analyzed using logistic regression after eliminating healthy subjects from the analysis and turning the “Severity” variable into a binary one, which indicates if the patient needs treatment or intensive care.

The choice of logistical regression analysis was dictated by the nature of the response variable, which is binary. Logistic regression analysis is the only statistical method that allows testing for the dependency of a binary variable, modeled as the probability of occurrence for a given event (i.e., contamination with COVID-19 and, respectively, a severe infection) on a set of potentially influencing factors of different statistical nature (binary, categorical, etc.) [100].

Provided that the difference in the odds ratios is not the same for the different levels of the predictors considered in this study, absolutely all independent variables were treated as categorical in all analyses. 

Model selection was performed with the intention of finding a model that includes only predictors with a significant impact on the response when acting together. It was performed using the approach proposed by Collett [101], which consists of: (1) performing univariate regressions with each predictor separately, (2) building a first multivariable model with all predictors if their *p*-values in (1) were below 0.2; (3) performing a backward elimination with the model obtained in (2) by eliminating the variable with the least influence on the response (indicated by the highest *p*-value) until all variables in the model have a statistically significant influence on the response. The fourth (4) procedure attempts to re-include variables not retained at the end of (3) in order to check if the model quality can be improved.

For all statistical analyses, the level of significance was 0.05. However, since the sample size was not very large, the presentation of results also includes results significant at the 0.1 level. In this case, additional data would probably have led to significant results.

## 3. Results

### 3.1. Descriptive Statistics

For a total of 1462 distributed questionnaires, we received 1125 responses: 753 responses (67.0%) were made in French, 225 (20.0%) in Arabic, 135 (12.0%) in English, and finally 12 (1.0%) in Spanish (Figure 1). The respondents were mainly French-speaking, which is conditioned by the previous geopolitical situation and the fact that French is practically the mother tongue of citizens of the majority of the countries in which the survey was conducted. In addition, the borders between North African and European countries had been mostly closed since the start of the COVID-19 pandemic. However, we certainly cannot exclude the possibility that immigrants participated in the study [102].

Of 1125 respondents, 332 were COVID-19 positive, 130 required home-based treatment, and 14 required intensive care (Table 1).

### 3.2. Data Analysis

Table 2 presents an overall image of analyses looking at the influence of the considered predictors on infection with COVID-19 and its severity. For each of the two, the table displays three models in separate columns. The first one looks at the individual relationship between infection with COVID-19 and its severity with each predictor. The second column presents a “full model”, ascertaining the simultaneous influence of all predictors on the infection with COVID-19 and its severity. Finally, the third column presents a “prediction model”, resulting from the full one and containing only predictors with a significant influence on infection with COVID-19 and its severity when considered simultaneously. The table clearly shows that age, sports, and health status are significantly associated with infection with COVID-19, whereas tobacco and protection are significantly associated with its severity.

Since not all factors in Table 3 were found to exert a statistically significant influence on having been affected or not by COVID-19, model selection was run in order to identify a model where all factors significantly affect the probability of having been affected or not by COVID-19. The model resulted by eliminating, in this order, the variables tobacco, gender, education, blood, and ethnicity. The resulting model, significant at *p* < 0.05, is displayed in Table 4.

## 4. Discussion

### 4.1. Significance of the Results and Health Recommendations

In this cross-sectional study, we evaluated the association between innate and lifestyle-related factors on the rate and severity of COVID-19. The main findings are as follows: (1) age, physical activity, and health status were found to have a significant influence on the infection of COVID-19; (2) observance of preventive measures and tobacco consumption had a significant influence on the severity of infection of COVID-19.

We confirmed the significant influence of the following factors on the possibility of infection: age, sports activity, and, to a lesser degree, health status. We could thus interpret the factors influencing the rate of infection by the fact that the older the person is, the more the individual will be subject to the infection; this was confirmed by the positive wideness of the estimated parameter of the logistic regression, thus agreeing with previous studies [30,32]. With regard to physical activity, our results showed that the more active the individual, the less likely he is to be infected with the virus; this was also confirmed by the positive wideness of the estimated parameter of the logistic regression, agreeing with previous studies [25,27]. For the health status, the estimated parameter indicates that the more the individual’s sensitivity increases, the more the individual will be subject to infection, confirming the results of previous studies concerning immunity [39]. Our results indicate that compliance with preventive measures is essential. Hence, the need to emphasize the observance of preventive measures (wearing masks, quarantine, remote working, hygiene, vaccination, etc.) becomes obvious. The problem is that the average population thinks that after being vaccinated, it is no longer mandatory to apply preventive measures [103]. It is true that previous studies have shown that vaccination of healthcare workers is associated with a substantial reduction in infections [104] and that one dose of vaccine reduces the potential for transmission by 61.0% [105]; however, the possibility of being infected still exists. On this matter, we encourage media and public health organizations to supply the public accurately with appropriate facts [106].

In the same vein as for the probability of infection, severity seems to increase with the intensity of smoking. In other words, the more severe infections will lead, in most cases, to hospitalization, thus confirming the logic of previous work [69,70]. Concerning protective measures, the logistic regression showed that the more the individual was strict in his application of the preventive measures promoted by the WHO and CDC [58], the less likely he would have a severe infection if he were infected. This contradicts a previous online questionnaire observational study that demonstrated that the use of protective measures was not associated with symptom severity of COVID-19 [107]. Another South Korean study showed that preventive measures had a negative influence on cardiometabolic profiles in subjects with metabolic impairments [108]. To our knowledge, there has not been any work that has demonstrated the positive effect of preventive measures on COVID-19. In contrast, a study in Pakistan showed that the application of protective measures dramatically reduces the risk of other respiratory diseases [109]. Furthermore, a recent Japanese study focused only on wearing a mask as a measure; they found that for a sample of 820 mask users, they were infected at a rate of 0.4 times that of those who did not wear masks [110], which demonstrates the importance of this measure. Our findings provide novel insights into the role of personal protective measures on the spread of infection and symptom severity. Further study is warranted to confirm our findings.

Regarding age, since the beginning of this pandemic, COVID-19 (and specifically its severity) has always been associated with old individuals [111]. The need for intensive care increases with age among individuals older than 45 years [112]. Our findings confirmed the importance of this factor by going in the same direction as previous reviews and meta-analysis studies [113,114].

Concerning alcohol, the situation is a little paradoxical. Our findings show that alcohol consumption has no significant influence on the probability of infection and its severity. However, McClain et al. [115] found the opposite because long-term drinking is closely related to zinc and vitamin deficiency. Hikida et al. [116] also found out that alcohol probably increases the risk of exposure to and morbidity from COVID-19. In addition, according to the same study, alcohol consumption is associated with prosecution and having regrets, which can increase the severity of COVID-19. González-Reimers et al. [117] discussed the fact that it has already been proven that various chronic disorders may increase the risk of pneumonia (and, by extrapolation, COVID-19) severity. Precisely for severity, a Chinese study performed on more than 1500 patients claimed that the severity of infection was greater in patients who consumed alcohol regularly or with a history of consumption of alcohol [118]. At the same time, a recent review article also confirmed that alcohol consumption adds vulnerability to patients and leads to an increased severity of COVID-19 and other long-term health problems [119]. Nevertheless, others remain a little less categorical, leaving doubt by pointing out the influence of several factors at the same time as well as the need for further studies [120,121].

For tobacco, the results of our work are quite logical because it is more associated with the severity of infection compared to infection. Indeed, non-smokers are obviously not immune to the virus; however, smoking patients are vulnerable to severe COVID-19 [122]. This perfectly confirms the findings of previous studies [123,124].

In general, the measures must be subject to a special focus, especially for sensitive sectors, citing as an example the education sector, where the application of measures was investigated in a few articles [125,126]. However, the most striking example remains, as stated previously, the health sector [127,128]. We think that it is important for each individual to educate himself about his own health on a regular basis. This issue remains debatable as, in many countries, the relationship between educational disparities and socioeconomic status is discussed a lot without reaching a final answer [129,130].

Variability in immune system components (innate) is a major contributor to the heterogeneous disease courses noticed for SARS-CoV-2 [131]. The immune response differs from one person to another and should not be relied on in any way to provide the still unknown character of several mechanisms [132]. A continuous effort to understand all mechanisms, especially those concerning host–virus interactions, is essential to overcome the pandemic [133]. In our study, a healthy state is associated with a reduced probability of contracting the virus, which can demonstrate the importance of innate immunity; however, in no case should we base our recommendations totally on this. Nevertheless, we cannot limit human health solely to these factors, and the effectiveness of vaccines plays the most important role [134].

Finally, our original research goal was to analyze the influence of two groups of factors and their influence on COVID-19 to reduce the spread of the virus. These findings also agree with the WHO reports, which have constantly repeated the various protective measures to be taken in order to avoid any possible infection and thus indirectly limit the spread of the virus [6]. Additionally, this study is one of only a few going against the idea that a part of the world population is automatically condemned to infection because of their ethnicity, place of residence, etc. Our study is meant to be encouraging by showing that each person can determine her/his destiny almost entirely as long as a healthy lifestyle is observed and precautions are taken; it is also about advising people to self-educate. This can only have positive effects, even in the long term.

### 4.2. Importance of the Study

The originality of this study comes from the fact that it is the only international study providing more information on the African and Mediterranean areas, carried out in four different languages. Indeed, the majority of previous studies have used one language, for example, French [135,136], Spanish [137], or Arabic [138]. Some studies used two languages, such as English and Hindi [139,140], for example, while four languages were used in the present study. The number of respondents (N = 1125) is quite high and heterogeneous compared to almost all questionnaires presented in previous studies; most studies had around between 300 and 500 responses [141,142,143]. Moreover, the vast majority of publications focused on a particular sector [144], a particular region [145], or a well-targeted population [146].

In addition, previous studies dealt with a single characteristic or group of characteristics [147,148], while the present study includes 13 items based on a substantial literature review. These items address two groups of characteristics, comparing the most significant ones. Everything was simplified as much as possible in order to reach all categories of the world population and collect more information applicable to the vast majority so that we can encourage them to adopt the right actions to prevent the infection and, ultimately, limit the spread of the disease. Moreover, the ranked influence of COVID-19 infection drivers has never been evaluated before. Any additional findings are important, provided their possible contribution to the fight against this pandemic.

### 4.3. Methodological Limitations

Our analysis was limited to a number of factors considered relevant based on the literature review and which could be ascertained using an online questionnaire. However, different studies also pointed out a number of other potential risk factors.

Objectively, it is an almost impossible challenge to know exactly the factors responsible for the infection and transmission of COVID-19. Sources may be incomplete; apart from the factors discussed previously, even meteorological ones have been considered a potential explanation [149]. A study in Korea demonstrated that the environment plays a significant role in the spread of COVID-19, but, like any factor, it may have also been impacted by various additional features [150]. Hence, further studies are needed to protect people from COVID-19 transmission, specifically on infection dynamics and the mode of transmission, e.g., cluster spaces, closed spaces, and indoor environments [151].

At the individual level, everyone must take the maximum possible precautions. It should also be remembered that no less than 10 reasons supporting airborne transmission were phrased recently by Greenhalgh et al. [152]. The long-term health consequences of COVID-19 remain unclear and continue to be studied [153]. Therefore, it is preferable to avoid any form of infection, even mild. Other factors that we do not necessarily think about and which may be important are wastewater treatment and disinfection strategies with chemical products; these demonstrate that we must consider an incalculable number of factors [154].

We can also mention certain biases concerning the sample. The majority of respondents were young intellectuals from the Arab world (Table 1) and certainly Muslim; consequently, tobacco and alcohol remain taboo subjects, so it is possible that some respondents did not tell the truth about these characteristics and our results could be biased; here are some noticed biases:(i).Integrating vaccination as a protective measure rather than as a factor in its own right when carrying out this sampling was not a limiting factor; time has shown that this is the case now; it was impossible at the time of developing the study, given the non-democratization of vaccinations at this time. For example, vaccination started in France on 27 December 2020 [155], on 29 December 2020 in Argentina [156], on 19 January 2021 in India [157], and on 10 February, Algeria [158].(ii).There could also be biases among social media users. For example, only 36% of Facebook users are over 35 years old [159].(iii).The same remark is valid regarding the languages. More than 60% of our sample answered the study in French, which could suggest that they are mainly located in the MENA region and French-speaking Europe or the possible presence of migrants.

The reliability of the statistical methodology is confirmed by the significance of all statistical models (Table 7), indicating that in all models, the predictors have, altogether, a significant effect on the response variable (infection and its severity). There is a slight difference with respect to the full model used to look at the severity of infection; two tests found it significant or marginally significant, and only one test found it not significant. While there is no power test associated with the logistic regression analysis, the overall significance of the statistical model is also an indication of the power of tests to identify significant associations between the response variables and associated risk factors.

### 4.4. Perspectives for Future Research

Undoubtedly, this study could not explicitly pretend to yield results that are able to stop this pandemic. In general, making predictions about this virus, which remains unknown, is impossible. Therefore, any additional information may prove useful. We consider that each study on this topic contributes in its own way to fighting indirectly against the pandemic crisis we are currently experiencing. Previously, we have enumerated a certain number of limitations in this work, including the number of answers received to the questionnaire. While the number of answers is acceptable, more answers could help obtain more representative results over a longer period and increase their statistical significance. Future studies should seek to use a larger and even more diverse sample and include additional recruitment strategies. For example, future studies should include in their sample information provided by parents of children under 18 years old, as they are also a group of high concern given the target specificity of the new virus variants. Furthermore, as mentioned with respect to the limitations of the study, a stronger focus on vaccination could be an excellent topic for future research, given the fact that as of May 2022, more than 67.6% of the world’s population has already been vaccinated [160], which allows the consideration of the specific inclusion of vaccination in future international studies. It could also be interesting to develop questionnaires using more languages (including Chinese, Hindi, Russian, and Portuguese, for example).

## 5. Conclusions

In this study, we found that health status, age, and physical activity significantly influence infection (*p* < 0.05). Not respecting preventive measures and smoking tobacco significantly aggravate its severity (*p* < 0.05). It would also appear that no matter where an individual is, what his/her age is, and everything that characterizes the person from birth, these are not, ultimately, the main characteristics to be taken into consideration concerning COVID-19 infection (and severity). Despite the limitations, based on our results, this is the first study that demonstrates that it is much more the lifestyle of the individual that decides a potential infection and its severity. Therefore, we strongly encourage the world population to adopt a healthy lifestyle: no alcohol, no tobacco, or at least reducing their consumption; eat healthily, sleep well, and practice physical activity regularly, in addition to vaccination, specifically for medical staff and people at risk. Finally, the main point not to be neglected will inevitably remain the issue of observing the protective measures promoted by the WHO and CDC since the epidemic started. Further studies remain necessary to help fight against the constantly evolving epidemiological situation. In the end, we hope that this positive message that we deliver (supported by results) can influence the behavior of people while waiting for the health crisis to subside and for the world to return to normality.

## Figures and Tables

**Figure 1 healthcare-10-01341-f001:**
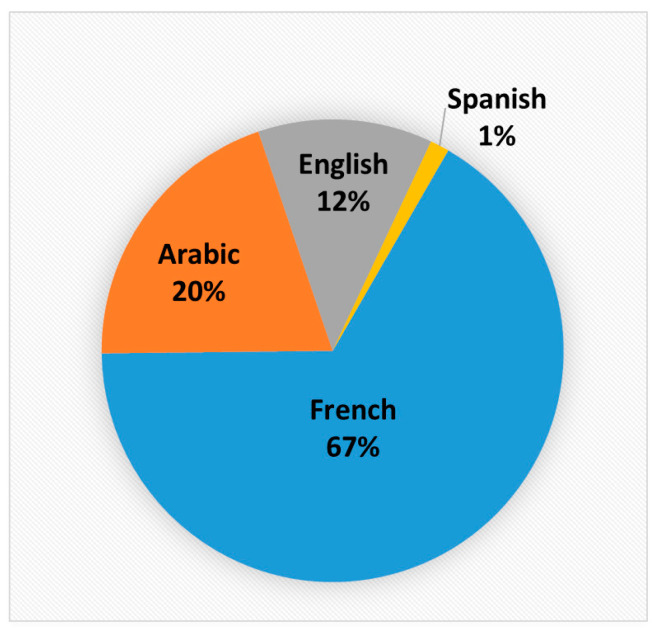
Representation of received answers.

**Table 1 healthcare-10-01341-t001:** Characteristics of the sample study.

Characteristics	Answers	Sample	Percentage %
Continent of residence	Africa	963	85.6%
	Europe	83	7.4%
	North America	19	1.7%
	South America	10	0.9%
	Asia	48	4.3%
	Oceania	2	0.2%
Ethnic origin	Not precise	13	1.2%
	Other	98	8.7%
	African/Afro-American	126	11.2%
	Caucasian	61	5.4%
	Arabic	794	70.6%
	Asian	22	2.0%
	Latino	11	1.0%
Gender	not precise	7	0.6%
	Male	394	35.0%
	Female	721	64.1%
	Other	3	0.3%
Age	18–30 years	727	64.6%
	31–45	315	28.0%
	46–59	60	5.3%
	60	23	2.0%
Blood group	not precise	13	1.2%
	A+	344	30.6%
	A−	21	1.9%
	B+	163	14.5%
	B−	17	1.5%
	AB+	56	5.0%
	AB−	8	0.7%
	O+	471	41.9%
	O−	32	2.8%
Educational attainment	Not precise	0	0.0%
	No study or primary	67	6.0%
	Middle or secondary	236	21.0%
	University or post-university	820	73.0%
	Little or no activity	557	49.6%
Sports activity	Moderate	451	40.2%
	Very active	114	10.2%
	Resistant (little or no flu/colds…)	730	65.3%
Health status	Moderately sensitive (regularly subject to flu/colds…)	331	29.6%
	Very sensitive (suffering from chronic disease(s) or others)	57	5.1%
	No	962	85.7%
Tobacco use	Occasionally	83	7.4%
	Frequently	78	6.9%
	No	1025	91.4%
Alcohol Consumption	Occasionally	81	7.2%
	Frequently	16	1.4%
	Not at all	65	5.8%
Observance of protective measures	Medium application	649	57.7%
	Strict application	411	36.5%
Infection with COVID-19	No	790	70.4%
	Yes	332	29.6%
	Healthy (no infection)	802	71.3%
Infection state	Low (no special care)	179	15.9%
	Treatment and /or care	130	11.6%
	Intensive care	14	1.2%

**Table 2 healthcare-10-01341-t002:** Analysis of the influence of considered predictors on infection with COVID-19 and its severity. The table describes the influence of a separate model (column labeled “Univariate”), a simultaneous model (column labeled “Full model”), and a final model resulting from the selection, including only predictors with a statistically significant simultaneous influence (column labeled “Selected model”). Bold values indicate a statistically significant impact (*p* ≤ 0.05), and those in *italics* have a marginally significant impact (0.05 < *p* ≤ 0.1).

Influence (Model)	Dependent Variable					
	Infection			Severity		
	Univariate	Full Model	Selected Model	Univariate	Full Model	Selected Model
Continent	0.3949	*0.0891*		0.2074	*0.0888*	
Ethnicity	0.3217	0.1553		0.9307	0.6429	
Gender	0.1654	0.7951		0.1088	0.7922	
Age	**0.0027**	**0.0185**	**0.0019**	0.4958	0.6296	
Blood	*0.0667*	0.1869		0.4963	0.4590	
Education	0.2661	0.5919		0.6485	0.6156	
Sports	**0.0009**	**0.0010**	**0.0005**	0.1478	0.2808	
Health	**0.0032**	*0.0919*	**0.0118**	*0.0723*	0.1300	
Tobacco	*0.0934*	0.6743		**0.0164**	0.3364	**0.0370**
Alcohol	0.1483	0.7615		0.1589	0.3068	
Protection	0.4519	0.4083		**0.0004**	**0.0042**	**0.0015**

Having been affected or not by COVID-19 was analyzed via logistic regression, the results of which are presented in Table 3. The model was overall significant (*p* < 0.0001). Logistic regression demonstrated that age, sports activity, and health status showed significant impacts on infection of COVID-19.

**Table 3 healthcare-10-01341-t003:** Results of logistic regression showing the factors determining whether a subject has been affected by COVID-19 or not. Bold values indicate a statistically significant impact (*p* ≤ 0.05), and those in *italics* indicate a marginally significant impact (0.05 < *p* ≤ 0.1). The model includes all analyzed factors.

Variable	DF	Wald Chi-Square	*p*-Value	Level	OR Estimate	95% Lower Wald Confidence Limit	95% Upper Wald Confidence Limit
*Continent*	*3*	*6.5147*	*0.0891*	*Africa* vs. *North America*	*0.355*	*0.013*	*9.687*
				*Asia* vs. *North America*	*0.691*	*0.014*	*34.708*
				*Europe* vs. *North America*	*2.781*	*0.083*	*92.650*
Ethnicity	4	6.6549	0.1553	Other vs. Asian	2.790	0.222	35.031
				African/Afro-American vs. Other	2.198	0.154	31.419
				Caucasian vs. Asian	9.962	0.571	173.918
				Arabic vs. Asian	4.956	0.369	66.604
Gender	1	0.0674	0.7951	Male vs. Female	1.089	0.571	2.076
**Age**	**3**	**10.0082**	**0.0185**	**18–30 vs. 60+**	**13.133**	**1.238**	**139.335**
				**31–45 vs. 60+**	**4.112**	**0.413**	**40.952**
				**46–59 vs. 60+**	**3.485**	**0.293**	**41.460**
Blood	7	10.0299	0.1869	A+ vs. O−	3.081	0.638	14.887
				A− vs. O−	<0.001 *	<0.001 *	>999.999 *
				B+ vs. O−	1.192	0.224	6.338
				B− vs. O−	4.288	0.125	147.175
				AB+ vs. O−	9.939	1.106	89.323
				AB− vs. O−	0.474	0.010	22.533
				O+ vs. O−	2.720	0.569	12.987
Education	2	1.0488	0.5919	No study or primary vs. University/post-university	1.401	0.514	3.820
				Middle/secondary vs. University/post-university	1.535	0.597	3.947
**Sports**	**2**	**13.7657**	**0.0010**	**Little or no activity vs. Very active**	**2.729**	**0.978**	**7.620**
				**Moderate vs. Very active**	**0.941**	**0.346**	**2.559**
*Health*	*2*	*4.7731*	*0.0919*	*Resistant* vs. *Very sensitive*	*4.078*	*1.069*	*15.559*
				*Moderately sensitive* vs. *Very sensitive*	*3.007*	*0.769*	*11.763*
Tobacco	2	0.7881	0.6743	No vs. Frequently	0.647	0.191	2.189
				Occasionally vs. Frequently	0.531	0.131	2.155
Alcohol	2	0.5448	0.7615	No vs. Frequently	0.683	0.077	6.035
				Occasionally vs. Frequently	0.460	0.041	5.129
Protection	2	1.7915	0.4083	Not at all vs. Strict application	1.629	0.546	4.859
				Medium application vs. Strict application	1.462	0.815	2.622

* The estimates are unusually large because, in reality, they exceed the program thresholds. This happens because of the very low sample size for specific levels (groups). While some were eliminated from the analysis (e.g., Gender—3 “Other” values, Continent—2 “Oceania” values, Health—1 “0 = Healthy” value, Continent—10 “4 = South America” values, and Ethnicity—11 “6 = Latino” values), it did not make sense to eliminate a blood group for medical reasons.

**Table 4 healthcare-10-01341-t004:** Results of the logistic regression showing the factors determining in a statistically significant way whether a subject has been affected by COVID-19 or not. All factors had a statistically significant impact (*p* ≤ 0.05). Some levels were eliminated from the analysis (e.g., Gender—3 “Other” values, Continent—2 “Oceania” values, Health—1 “0 = Healthy” value, Continent—10 “4 = South America” values, and Ethnicity—11 “6 = Latino” values), but it did not make sense to eliminate a blood group for medical reasons.

Variable	DF	Wald Chi-Square	*p*-Value	Level	OR Estimate	95% Lower Wald Confidence Limit	95% Upper Wald Confidence Limit
Age	3	14.8694	0.0019	18–30 vs. 60+	10.841	1.247	94.234
				31–45 vs. 60+	4.707	0.531	41.732
				46–59 vs. 60+	3.940	0.377	41.199
Sports	2	15.3573	0.0005	Little or no activity vs. Very active	2.820	1.192	6.668
				Moderate vs. Very active	1.037	0.448	2.400
Health	3	8.8880	0.0118	Resistant vs. Very sensitive	5.158	1.587	16.763
				Moderately sensitive vs. Very sensitive	3.384	1.010	11.337

The dependence of the severity of infection on the potential risk factors investigated was analyzed using logistic regression. The results are presented in Table 5. The model was overall significant (*p* = 0.0201). The table indicates that the observance of protective measures was significantly associated with the severity of infection, and the continent of origin had a marginally significant influence on it.

**Table 5 healthcare-10-01341-t005:** Results of logistic regression showing the factors determining whether a subject has been severely affected by COVID-19 or not. Variables with names written in bold front have a statistically significant impact, and those in *italics have* a marginally significant impact (0.05 < *p* ≤ 0.1). The model includes all analyzed factors. Some levels were eliminated from the analysis (e.g., Gender—3 “Other” values, Continent—2 “Oceania” values, Health—1 “0 = Healthy” value, Continent—10 “4 = South America” values, and Ethnicity—11 “6 = Latino” values), but it did not make sense to eliminate a blood group for medical reasons.

Variable	DF	Wald Chi-Square	*p*-Value	Level	OR Estimate	95% Lower Wald Confidence Limit	95% Upper Wald Confidence Limit
*Continent*	*3*	*6.5229*	*0.0888*	*Africa* vs. *North America*	*0.531*	*0.156*	*1.803*
				*Asia* vs. *North America*	*1.339*	*0.303*	*5.916*
				*Europe* vs. *North America*	*0.966*	*0.255*	*3.654*
Ethnicity	4	2.5095	0.6429	Other vs. Asian	1.873	0.593	5.916
				African/Afro-American vs. Other	1.568	0.497	4.953
				Caucasian vs. Asian	1.167	0.336	4.050
				Arabic vs. Asian	1.756	0.582	5.299
Gender	1	0.0694	0.7922	Male vs. Female	0.957	0.690	1.327
Age	3	1.7330	0.6296	18–30 vs. 60+	1.450	0.539	3.899
				31–45 vs. 60+	1.199	0.455	3.162
				46–59 vs. 60+	0.986	0.330	2.949
Blood	7	6.7163	0.4590	A+ vs. O−	1.465	0.643	3.341
				A− vs. O−	5.754	1.067	31.035
				B+ vs. O−	1.655	0.695	3.944
				B− vs. O−	3.469	0.624	19.265
				AB+ vs. O−	2.032	0.734	5.626
				AB− vs. O−	1.575	0.241	10.299
				O+ vs. O−	1.422	0.630	3.209
Education	2	0.9703	0.6156	No study or primary vs. University/post-university	0.946	0.538	1.664
				Middle/secondary vs. University/post-university	1.250	0.767	2.035
Sports	2	2.5403	0.2808	Little or no activity vs. Very active	0.968	0.585	1.603
				Moderate vs. Very active	0.777	0.470	1.283
Health	2	4.0803	0.1300	Resistant vs. Very sensitive	1.273	0.691	2.345
				Moderately sensitive vs. Very sensitive	0.943	0.502	1.772
Tobacco	2	2.1788	0.3364	No vs. Frequently	0.985	0.550	1.765
				Occasionally vs. Frequently	0.673	0.331	1.368
Alcohol	2	2.3629	0.3068	No vs. Frequently	1.864	0.550	6.324
				Occasionally vs. Frequently	1.243	0.334	4.631
**Protection**	**2**	**10.9263**	**0.0042**	**Not at all vs. Strict application**	**0.387**	**0.210**	**0.715**
				**Medium application vs. Strict application**	**0.700**	**0.517**	**0.947**

Since not all factors in Table 5 were found to exert a statistically significant influence on the severity of COVID-19 infection, model selection was run in order to identify a model where all factors had a significant influence. The model resulted by eliminating, in this order, the variables gender, education, ethnicity, tobacco, blood, sports, continent, and age. The resulting model, significant at *p* < 0.05, is displayed in Table 6. The table indicates that the observance of protective measures and smoking were significantly associated with the severity of infection.

**Table 6 healthcare-10-01341-t006:** Results of the logistic regression showing the factors determining (in a statistically significant manner) whether a subject has been severely affected by COVID-19 or not. All factors have a statistically significant impact (*p* ≤ 0.05). Some levels were eliminated from the analysis (e.g., Gender—3 “Other” values, Continent—2 “Oceania” values, Health—1 “0 = Healthy” value, Continent—10 “4 = South America” values, and Ethnicity—11 “6 = Latino” values), but it did not make sense to eliminate a blood group for medical reasons.

Variable	DF	Wald Chi-Square	*p*-Value	Level	OR Estimate	95% Lower Wald Confidence Limit	95% Upper Wald Confidence Limit
Tobacco	2	6.5963	0.0370	No vs. Frequently	0.983	0.589	1.641
				Occasionally vs. Frequently	0.539	0.280	1.040
Protection	2	13.0085	0.0015	Not at all vs. Strict application	0.382	0.220	0.663
				Medium application vs. Strict application	0.725	0.547	0.961

**Table 7 healthcare-10-01341-t007:** Power of the logistic regression analysis models used in the study. The table displays the value of the overall tests, testing whether the entire set of predictors has a significant effect on the response variable. Tests with names written in **bold** font were found significant (*p* ≤ 0.05), and those in *italics* were marginally significant (0.05 < *p* ≤ 0.1).

Significance Test	Infection		Severity	
	Full Model	Selected Model	Full Model	Selected Model
**Likelihood Ratio**	**<0.0001**	**<0.0001**	**0.0446**	**<0.0001**
**Score**	**0.0007**	**<0.0001**	*0.0605*	**0.0291**
**Wald**	**0.0263**	**<0.0001**	0.1078	**0.0180**

## Data Availability

The data generated and/or analyzed in the current study are available from the corresponding author upon reasonable request.

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
