# Peer review of "The Importance of Behavioral and Native Factors on COVID-19 Infection and Severity: Insights from a Preliminary Cross-Sectional Study"

_healthcare, 2022, doi:10.3390/healthcare10071341_

Round 1

Reviewer 1 Report

Title: I suggest to be reformulated. 86% of the respondents came from Africa and 71% Arabs so it is difficult to draw conclusions about the impact of cultural and ethnic differences on infection and severity of disease.

Abstract:  line 39-41 I recommend to detail your result -Better physical activity and better health status significantly reduced the possiblity of infection while...older age and so on...... the severity of infection was negatively associated with acceptance (adherenece, respect) of protective mesures....

Introduction: line: higher spreading nature is quite uncommon term (I suggest basic reproductive number or transmissibility)

Line 95: Please, explain how do you choose this 64 studies out of the 170 found to be relevant?

Line 130: Reformulate this sentence- susceptibility to SARS-CoV-2 is universal, but older age (and some other factors) are associated  with disease severity.

Line 209: Provide more information on other population-dependent antigens to save readers from reading references

Line 287: Is it realy underreporting rather than undertesting responsible to  great variation in infection rates across the globe?

Methodology

Line 345: It seems to me to be difficult to asses the resistance (and subsequenty immunity ) to COVID-19 and all your subsequent conclusions by selfreported health status as resistant, moderate sensitive, very sensitive). May you provide additional references for this desicion?

Discussion:

Line 503: contamination possibility is not a good choice of word to say possibility of be infected or something like that.

Line 523: Older age has been associated with disease severity more than infection

Line 531: It seems that this is not correct references for this conclusion. I found it and read it . I suggest to correct it and check the position of all references in the manuscript.

Line 564: Please, explain why do you mention relationship between vaccines, pollution  and their consequences?

Line 566: Discussion related the sources of information is here out of the box. (No assessed in this study)

Author Response

We want to express our sincere gratitude to Reviewer #1 for the time dedicated to the review and the comprehensive, profound, and constructive remarks, which allowed us to improve the quality of our manuscript. The table attached presents in detail how to address each comment; the references are to the final line numbers of the revised article. In addition, the added or changed text of the manuscript was marked using “track changes” of Microsoft Word. We believe that, this paper will be cited frequently by other authors.

Reviewer 2 Report

I would like to thank you for the opportunity to review this manuscript. The paper aimed to understand the factors influencing the rate and severity of COVID-19 such as innate and lifestyle-related ones. I appreciate the importance of this work, particularly as it captures community differences through a multi-language survey.

I am glad the authors brought this to the paper. The paper is presented well throughout, and the discussions are clear and appropriate given the study’s aims and rationale. I believe this manuscript is at a standard for publication.

Author Response

We want to express our sincere gratitude to Reviewer #2 for the time dedicated to the review and the comprehensive, profound, and constructive remarks, which allowed us to improve the quality of our manuscript. The table attached presents in detail how to address each comment; the references are to the final line numbers of the revised article. In addition, the added or changed text of the manuscript was marked using “track changes” of Microsoft Word. We believe that, this paper will be cited frequently by other authors.

Reviewer 3 Report

After conducting a narrative review of literature, the authors investigated, by means of an anonymous questionnaire written in different languages and administered online, how innate factors and lifestyle can influence the virus spread and the development of SARS Cov 2 disease.
Overall, the work is interesting and it highlights the importance of lifestyle and a low influence of innate factors. The questionnaire is well structured, according to a good methodology, the statistical analysis is appropriate and the bibliography exhaustive.
In consideration to the subject, the study would have been more relevant if the sample had been larger. It would also have been useful to know the number of questionnaires distributed against those which returned completed. This would have made it possible to understand the degree of interest aroused by the survey.

Author Response

We want to express our sincere gratitude to Reviewer #3 for the time dedicated to the review and the comprehensive, profound, and constructive remarks, which allowed us to improve the quality of our manuscript. The table attached presents in detail how to address each comment; the references are to the final line numbers of the revised article. In addition, the added or changed text of the manuscript was marked using “track changes” of Microsoft Word. We believe that, this paper will be cited frequently by other authors.

Reviewer 4 Report

Abstract:

Line 39: The significant factors included age, physical activity and health status all found to have a significant influence on the infection (p < 0.05).

-Positive influence or negative influence, please clarify.

Line 41. The severity of infection was associated with preventive measures and tobacco (p < 0.05).

-Reorganize this statement. What does tobacco mean here?

Line 74: Previously, the published studies were mainly interested in the influence of COVID-19 on the changes of lifestyle and not the opposite 

-I am not sure if I clearly understand what the authors are trying to say here.

-The introduction section can be better organized.

-Overall, an interesting study that focuses on the lifestyle of the individual with disease severity. Multiple grammatical errors need to be corrected. 

Author Response

We want to express our sincere gratitude to Reviewer #4 for the time dedicated to the review and the comprehensive, profound, and constructive remarks, which allowed us to improve the quality of our manuscript. The table attached presents in detail how to address each comment; the references are to the final line numbers of the revised article. In addition, the added or changed text of the manuscript was marked using “track changes” of Microsoft Word. We believe that, this paper will be cited frequently by other authors.

Round 2

Reviewer 4 Report

The authors have made appropriate changes recommended by reviewers. 

This manuscript is a resubmission of an earlier submission. The following is a list of the peer review reports and author responses from that submission.

Round 1

Reviewer 1 Report

Thanks for allowing me to review this highly interesting paper. Here are my main comments:

  1. You should write in the abstract that you did a literature review to define your variables of interest.
  2. Moreover, please be as precise as possible about this literature review. You wrote on section 1.2 (see page 3, lines 100-102) that "Keywords such as "coronavirus", "COVID-19", (...), "alcohol", "lifestyle", etc. were included". Please remove the "etc." and write exactly what you included. If it is too long, you can put the list as as appendix. As a reader, I would like to know exactly what you looked for.
  3. All the abstract and results focus on p-values. Personally I don't care if you have p=0.049 or p=10^(-16). You "just" need to have a bigger sample in order to get significant p-values. The p-value itself is not interesting, you need to report effect sizes. When you report logistic regressions' results, please write Odds Ratios, 95% confidence intervals and p-values. When you report linear regressions' results, please write coefficients, 95% CI and p-values. Don't write p-values alone, this is not interesting.
  4. Reporting in Tables 2 and 3 can be improved. Sum of squares, F-values are not interesting. Again, please report odds ratios, 95% CIs and p-values.
  5. About your model selection, I suggest the following procedure (see the book of David Collett called "Modelling Binary Data"):
    a) perform univariate regressions with each predictor.
    b) build a first multivariable model with all predictors such that p-values in a) was below 0.2
    c) proceed to backward elimination with the model obtained in (b).
    d) try again to include variables that are not retained at the end of (c), in order to check if you can improve model quality.
    Maybe this procedure will give you a different final model. At least it needs to work carefully at each step, and not simply push "backward elimination" button and take the result. You can then be more trustful about the obtained model, to ensure it is somehow the best model.

Minor comments:

  1. The first percentage about 332/1125 is wrong. You wrote 29.6% but it should be 29.5%. I did not verify the other numbers, but please do it.
  2. In general, please be consistent with those percentages (always one digit).

Reviewer 2 Report

The authors designed, tested, and implemented an international survey in hopes of finding if any behavioral factors are associated with greater risk of SARS-CoV-2 infection and subsequent severity of disease. The use of an international survey is useful for broader applicability, and the results are helpful for identifying populations at greater risk.

MAJOR COMMENTS

  1. It appears in the data analyses that many of the ordered, categorical variables were treated as continuous. These include health status (severity of disease), tobacco use, alcohol consumption, and age group, among others. This can be problematic by assuming that the differences across categories are linear. For example, it assumes that the difference in log odds of having COVID-19 (from the logistic regression) between people who consume alcohol occasionally and people who do not consume alcohol is the same as the difference in log odds between people who consume alcohol frequently and people who consume alcohol occasionally. Further, it assumes that the difference in log odds between people who consume alcohol frequently and people who do not consume alcohol is double the previously mentioned difference. This is generally not an appropriate assumption, and the authors would benefit from treating all of these variables as categorical.

  1. It is important to show the parameter effect estimates from the logistic regression. This allows readers to see both direction and magnitudes of associations. Using p-values alone to conclude importance, especially relative importance among predictors, is not an ideal representation.

  1. The analyses of COVID-19 severity use ANCOVA, which assumes a continuous and normally distributed outcome. However, the outcome variable used is a four-category outcome, with one of the four categories, denoted “healthy” presumably indicated not having COVID-19. Treating these categories as continuous in the ANCOVA and the correlation analysis is problematic, and the inclusion of respondents who did not have COVID-19 makes interpretations odd. It would be beneficial to restrict analysis of severity of COVID-19 to only participants who had COVID-19.

MINOR COMMENTS

  1. It would be beneficial in the abstract and beginning of the discussion section to use effect estimates form analyses rather than only p-values when listing important predictors of COVID-19 incidence and severity. Generally, p-values reflect precision of the effect estimates more than their magnitude or relative importance to prediction.

  1. The vaccine info on lines 76-90 may need updating or a reference date (as of ___...) in case any information becomes out of date during the revision process.

  1. Section 1.2.4 discusses blood types. It appears that a blood group is denoted with the number zero rather than the letter O.

  1. Lines 175-178: Is there a location associated with this study for context? Impacts of race/ethnicity are likely to vary in different countries, so it is important to contextualize impacts of race/ethnicity to a global audience.

8.Does the survey question regarding COVID-19 status ask if the respondent currently has COVID-19 while taking the survey, has ever had COVID-19, or another interpretation?

  1. The estimates in Table 5 need to be contextualized. Do these represent the associations between the predictors and the severity outcome directly as described in the methods? If so, then it appears that health and alcohol have an inverse relationship with disease severity, which contradicts the correlation analysis and main text.

  1. Figure 2 appears to use a pyramid to represent relative importance of the predictors. From the logistic regression, it appears that this was only based on p-values. The p-values are representative of the precision of the effect estimate rather than the magnitude. A predictor can have a low p-value but also have a low (but precise) magnitude of association. Consistency is needed between the two pyramids, and the rationale for creating this hierarchy needs to be stated.

Reviewer 3 Report

The manuscript entitled “Association between Innate and Lifestyle-related Characteristics on the Rate and Severity of COVID-19: an International Questionnaire-based Survey (ijerph-1463771)” reported that the individual behavior is mainly responsible for the spread of the virus based on an anonymous internet survey of 1125 respondents from six countries. The strength of this manuscript is a detailed introduction. However, the questionnaire item (especially protective measures against COVID-19, tobacco use, and alcohol consumption) and the cross-sectional study are not appropriate at all. The statistical analysis was incorrectly conducted, hence, the results and conclusion are incorrect.  

  1. Logistic regression should be used for the outcome variable (affected by COVID-19 or not). However, Tables 2 and 3 are based on ANCOVA.
  2. The severity of the state of infection with COVID-19 is not normally distributed, hence, ANCOVA is not appropriate (Table 4-5).

Reviewer 4 Report

The authors have performed a cross-sectional study using an online questionnaire to assess the association between lifestyle-related characteristics and incidence/severity of COVID-19. The manuscript is well-written and has an important insight to report. I have the following concerns.

  1. I would like to have more detailed discussion regarding the association between preventive measures and COVID-19 severity. Please describe the underlying mechanisms where preventive measures could contribute to not only COVID-19 infection but also COVID-19 severity.
  2. It would be lovely to have further discussion regarding the reason why tobacco did not show any significant influence on COVID-19 infection and severity.
  3. I was curious to know if the authors had the information of financial status.
  4. I would like to know which combination of factors is assumed to affect significantly COVID-19 infection and severity.
  5. The conclusion sounds a little general and leap. It can be improved by describing based on the findings in the current study. 

Round 2

Reviewer 1 Report

I first want to thank the authors for having considered my remarks and those from the other reviewers. I think the quality clearly improved. Removing ANCOVA, considering categorical variables, expressing ORs and CIs are really good. We are close to acceptance but I still have a few remarks:

  1. You removed the ANCOVA from your analyses, but the latter is still expressed at some places (in the abstract, line 34, or on page 13, lines 430-432). Please remove this.
  2. You should rephrase the definition of Severity on page 8, lines 372-376. You write it is a binary variable but then give four levels. It is not straightforward to understand that the first two levels are 0 and the two top levels are 1. Here is my suggestion:
    "(...) after turning "Severity" variable into a binary one, which indicates if the patient needs treatment or intensive care."
  3. Please change the column names in Table 2. You wrote "Separate / Simultaneous / Prediction model", but I would prefer "Univariate / Full model / Selected model". Indeed, your work did not cover the topic of prediction (which could be another paper!).
  4. Thank you very much for giving ORs and 95% CIs! However, the first one we see is OR=">999.999" and CI: <0.001 to >999.999. You have a lot of extremely high values of OR and extremely wide CIs. This is not nice at all and I wondered why this happen... I then realize that you always take the highest value of your predictor as the reference category. As an example, with Gender, you show results "1 vs. 3" and "2 vs. 3". This means "Male vs. Other" and then "Female vs. Other". But you don't show the comparison Male vs. Female, which is the only one interesting here, at least to my opinion. And since they are only 3 "Other"s, the ORs and CIs are horrible and not informative at all.
    In general, please redo the tables by showing the category 1 as the reference (i.e. 1 vs. 2, 1 vs. 3, 1 vs. 4, and so on). This will give much more interesting tables.

Reviewer 2 Report

Thank you for the revised manuscript. It is clear that improvements were made to the analyses, allowing more valid interpretations of results. Some smaller concerns remain in the revised version

1. I am assuming the model ORs of >999 represent an undefined OR rather than an extremely large OR. This should be addressed in the text, or, preferably, analyses should be modified so that defined ORs can be estimated. A potential solution would be to take greater care in selection of reference groups, as some reference groups (South America, other gender, for example) have very low sample sizes. A common practice is to select reference groups that lead to most useful interpretations or to select the most common value as the referent (such as using Africa as the referent continent).

2. In the results tables, using words rather than numbers to describe the ORs would be more helpful for easily understanding the comparisons being made.

3. The text states that analysis of disease severity was done using ANCOVA, but the tables indicate that logistic regression was used. This disagreement should be clarified.

4. While the conclusions of this study highlight some individual behavioral factors that can influence risk of COVID-19 infection and severity, the authors correctly acknowledged that it is impossible to have a perfect and complete picture of all risk factors. It seems a stretch to then state that individuals are entirely responsible for their own COVID-19 risk (lines 538-540), since there still exist many factors beyond one’s own control. While it is helpful to state that some individual behaviors can help, the wording should be modified to avoid placing all responsibility on the individual.

5. Attention to English conventions is needed. Below are two examples with potential corrections:

-lines 72-73 lists drugs used, but is not a complete sentence. Consider: “Other notable examples of proposed remedies include Remdesivir …”

-lines 448-451 contain a run-on sentence followed by an incomplete sentence. Consider: “However, our findings show that the possibility of contamination is statistically significantly influenced by age, physical activity, and health status. All of these were found to have a significant influence on infection, as shown in Tables 2-4.”

Reviewer 3 Report

The manuscript entitled “Association between Innate and Lifestyle-related Characteristics on the Rate and Severity of COVID-19: an International Questionnaire-based Survey (ijerph-1463771)” reported that the individual behavior is mainly responsible for the spread of the virus based on anonymous internet survey of 1125 respondents from six countries. The statistical analysis was incorrectly conducted, hence, the results and conclusion are incorrect.  

  1. Useless information of Table 2. The strange result of Table 3 with OR>999.9. Table 4 and 5 is very difficult to read because we don’t know the meaning of the level of different variables.
  2. Discussion: not relative to the result at all.
  3. Abstract: useless.

Reviewer 4 Report

Thank you for the revised manuscript. Some improvements were found but I still have the following concerns in the newly conducted statistical analyses.

  1. Table 3 demonstrated ORs > 999.9 but these results do not seem make sense. Please consider re-analyzing or describing the appropriate explanations.
  2. Please explain the detailed methodology and the validity on the logistic regression analyses performed in Table 4 and Table 5. Also please explain the appropriate interpretation of these results of Table 4 and Table 5 in the result section. The authors also may want to explain how different the significance of each variables is between all the variables and how to compare the significance of these variables in different groups in Table 4 and Table 5.
  3. The abstract and the conclusion still sound like a general information. It would be beneficial by incorporating the novel findings in the current study into the abstract and the conclusion.